# CD45-Directed CAR-T Cells with CD45 Knockout Efficiently Kill Myeloid Leukemia and Lymphoma Cells In Vitro Even after Extended Culture

**DOI:** 10.3390/cancers16020334

**Published:** 2024-01-12

**Authors:** Maraike Harfmann, Tanja Schröder, Dawid Głów, Maximilian Jung, Almut Uhde, Nicolaus Kröger, Stefan Horn, Kristoffer Riecken, Boris Fehse, Francis A. Ayuk

**Affiliations:** 1Research Department Cell and Gene Therapy, Department of Stem Cell Transplantation, University Medical Centre Hamburg-Eppendorf (UKE), 20246 Hamburg, Germanya.uhde@uke.de (A.U.);; 2Department of Stem Cell Transplantation, University Medical Centre Hamburg-Eppendorf (UKE), 20246 Hamburg, Germany

**Keywords:** chimeric antigen receptor (CAR), CD45, CRISPR-Cas, cytotoxicity, leukemia, immunotherapy, conditioning

## Abstract

**Simple Summary:**

Chimeric antigen receptors (CARs) are used to recognize highly specific antigens (“mugshots”) to target immune effectors (“policemen”) against cancer cells. Whereas this new immunotherapy has already set novel standards in the treatment of some specific types of blood cancer, it has not yet been successful with most blood (and solid) cancers for different reasons. In this work, the authors investigated the possibility of using a specific antigen called CD45 as target for CAR therapies. CD45 stands out by being present on all blood cells and therefore represents a promising target in any type of blood cancer. However, immune cells themselves also harbor CD45 at their cell surface, which is expected to lead to either inactivity of the CAR or self-killing of immune cells. To avoid these problems the Hamburg group used a trick—they applied the CRISPR/Cas gene scissors and thus produced CD45-knockout (CD45^ko^) immune-effector cells. Even though CD45 had been supposed to be important for the functionality of immune-effector cells, the authors observed excellent survival, proliferation and killing activities of their CD45^ko^ CAR cells, independent of the targeted cancer-cell population. Their results thus provide initial proof-of-concept for the potential usefulness of CD45^ko^/CD45-CAR immune cells to target blood cancer.

**Abstract:**

Background: CAR-T cell therapy has shown impressive results and is now part of standard-of-care treatment of B-lineage malignancies, whereas the treatment of myeloid diseases has been limited by the lack of suitable targets. CD45 is expressed on almost all types of blood cells including myeloid leukemia cells, but not on non-hematopoietic tissue, making it a potential target for CAR-directed therapy. Because of its high expression on T and NK cells, fratricide is expected to hinder CD45CAR-mediated therapy. Due to its important roles in effector cell activation, signal transduction and cytotoxicity, CD45 knockout aimed at preventing fratricide in T and NK cells has been expected to lead to considerable functional impairment. Methods: CD45 knockout was established on T and NK cell lines using CRISPR/Cas9-RNPs and electroporation, and the successful protocol was transferred to primary T cells. A combined protocol was developed enabling CD45 knockout and retroviral transduction with a third-generation CAR targeting CD45 or CD19. The functionality of CD45^ko^ effector cells, CD45^ko^/CD45CAR-T and CD45^ko^/CD19CAR-T cells was studied using proliferation as well as short- and long-term cytotoxicity assays. Results: As expected, the introduction of a CD45-CAR into T cells resulted in potent fratricide that can be avoided by CD45 knockout. Unexpectedly, the latter had no negative impact on T- and NK-cell proliferation in vitro. Moreover, CD45^ko^/CD45CAR-T cells showed potent cytotoxicity against CD45-expressing AML and lymphoma cell lines in short-term and long-term co-culture assays. A pronounced cytotoxicity of CD45^ko^/CD45CAR-T cells was maintained even after four weeks of culture. In a further setup, we confirmed the conserved functionality of CD45^ko^ cells using a CD19-CAR. Again, the proliferation and cytotoxicity of CD45^ko^/CD19CAR-T cells showed no differences from those of their CD45-positive counterparts in vitro. Conclusions: We report the efficient production of highly and durably active CD45^ko^/CAR-T cells. CD45 knockout did not impair the functionality of CAR-T cells in vitro, irrespective of the target antigen. If their activity can be confirmed in vivo, CD45^ko^/CD45CAR-T cells might, for example, be useful as part of conditioning regimens prior to stem cell transplantation.

## 1. Background

Chimeric antigen receptor (CAR-) T-cell therapy is now part of standard care in the treatment of patients with acute lymphoblastic leukemia (ALL), B-cell lymphoma, and multiple myeloma [1,2,3,4,5,6]. This innovation has been enabled mainly by targeting B-cell lineage-specific cell surface antigens, namely CD19 and B-cell maturation antigen (BCMA). In contrast, the development of CAR-T cell therapy for myeloid diseases, especially acute myeloid leukemia (AML), has been hampered primarily by the absence of suitable AML-specific or at least AML-associated antigens. Indeed, the targeting of myeloid-lineage associated antigens such as CD33 and CD123 has been impeded by their expression in other tissues (CD33 in liver, CD123 in endothelial cells [7,8]), which raises issues of on-target off-tumor toxicity. 

CD45 is a type-I transmembrane protein with various isoforms found on almost all hematopoietic cells (except erythrocytes and platelets) and a regulator of signaling thresholds in immune cells [9]. It is expressed on 85–90% of ALL and almost all AML cells but not on non-hematopoietic tissues [10,11]. Targeting CD45 with CAR-based cell therapy may therefore in principle be employed for the treatment of both lymphoid and myeloid hematological diseases. However, so far, CD45 has not been explored for CAR-based therapy for several reasons. Owing to its high expression on T cells and NK cells, the introduction of CD45-directed CARs is expected to result in fast and widespread fratricide strongly impairing efficacy. Moreover, since CD45 is present on all nucleated hematopoietic cells, successful CD45-directed CAR-T cell therapy is anticipated to result in profound myeloablation. 

One potential approach to prevent fratricide is through CD45 knockout in effector cells prior to their transduction with a CD45-CAR. Highly efficient CD45 knockout (KO) in human T and NK cells was previously reported using “ready-to-go” ribonucleoprotein complexes consisting of synthetic sgRNA and recombinant Cas9, but the functionality of those cells was not studied [12,13]. Given the central role of CD45 in the regulation of cell activation, signal transduction, and cytotoxicity [14], a significant loss of function of immune effector cells is expected upon knockout of CD45.

In this study, we sought to generate CD45^ko^ CAR-effector cells and examine their functional properties in various in vitro assays. 

## 2. Methods

### 2.1. Cell Culture

Peripheral blood mononuclear cells (PBMCs) were isolated from buffy coats (BCs) via density gradient using Secoll Lymphocyte Separation Medium (Serana, Pessin, Germany). BCs were anonymized leftovers of whole-blood donations from healthy donors. They were kindly provided for research (with informed consent) by the Institute of Transfusion Medicine, University Medical Center Hamburg-Eppendorf, Hamburg, Germany. To obtain primary T cells, freshly isolated PBMCs were activated with T Cell TransAct for human T cells (Miltenyi Biotec, Bergisch Gladbach, Germany). Briefly, PBMCs were seeded in 6-well suspension cell culture plates at cell concentrations of 1.5 × 10^6^ per mL. T cell TransAct was added according to the manufacturer’s instructions at a 1:100 ratio. The primary T cells were cultured in TexMACS medium (Miltenyi Biotec) supplemented with 3% human serum (Merck, Darmstadt, Germany) and 200 U/mL hIL-2 (Proleukin S, Novartis, Basel, Switzerland). In the case of long-term culture, the activation step with T cell TransAct was repeated after 14 days.

Ramos (ATCC, CRL-1596), KHYG-1 (DSMZ, ACC-725), and PM-1 cells [15] were cultured in RPMI medium 1640 (Gibco/ThermoFisher Scientific, Schwerte, Germany) supplemented with penicillin-streptomycin (100 U/mL and 100 μg/mL; Gibco) and 2 mM L-glutamine (Gibco). The medium was further supplemented (1) with 25 mM HEPES (Gibco) and 13% heat-inactivated fetal calf serum (FCS) (Gibco) for Ramos, (2) with 20% FCS, 1% sodium-pyruvate (Gibco), 1% non-essential amino acids (NEAA, Gibco), and 200 IU/mL IL-2 for KHYG-1, and (3) with 10% FCS, 1% sodium pyruvate for PM-1 cells. HEK293T cells (ATCC CRL-3216) were kept in Dulbecco’s modified Eagle’s medium (DMEM Glutamax, Life Technologies, Carlsbad, CA, USA) supplemented with 10% FCS and penicillin-streptomycin (100 U/mL and 100 μg/mL). The GM-CSF-dependent human AML-cell line Nos-1FD4 (established in our lab, Horn et al., manuscript in preparation) was maintained in RPMI 1640 supplemented with 10% FCS, penicillin-streptomycin (100 U/mL and 100 μg/mL), 1% sodium pyruvate, 1% NEAA, 0.1% beta-mercaptoethanol (Gibco) and 5 ng/mL human GM-CSF (Peprotech/ThermoFisher Scientific). A variant carrying an FLT3-mutation (Nos-1FD4-ITD) after lentiviral transduction was cultivated without GM-CSF. Cells were incubated under standard conditions: 37 °C, 100% relative humidity, 5% CO_2_.

### 2.2. Cloning of the CD45-CAR and Retroviral Vector

Vector cloning was performed using established recombinant-DNA techniques, essentially as described [16]. The general structure of the CAR expression construct is shown in Figure 1 and largely follows previous work [17]. The used third-generation CAR contained the following elements: An anti-CD45 single-chain variable fragment (scFv) derived from the mouse anti-human CD45 antibody clone BC8 [18], spacer and CD28-derived transmembrane (TM) regions, and tandem CD28 and 4-1BB co-activation domains linked to the CD3ζ signaling domain. Two different spacer variants were used, a short CD28-derived spacer and a longer one derived from IgG4, including its hinge, CH2 and CH3 domain. DNA sequences were codon-optimized. The CAR was cloned into the γ-retroviral vector pRSF91.iB.pre*-Puro^+^ upstream of an encephalomyocarditis virus (EMCV) internal ribosomal entry site (IRES) and the coding region for the blue fluorescent protein mTagBFP, linked to the puromycin resistance PAC by a 2A peptide. After integration in the host cell genome, the CAR is expressed under control of the SFFV promoter.

### 2.3. Retroviral Transduction

Retroviral vector particles pseudotyped with the Gibbon-ape leukemia virus (GALV) envelope were produced by transiently transfecting HEK293T cells. To do so, we used the calcium phosphate method and titrated the obtained vector containing supernatants on HEK293T, both steps as described previously [21]. Titers of concentrated (by centrifugation) supernatants were in the range of 1–60 ×10^6^ transducing units per mL. 

Primary T lymphocytes were transduced on RetroNectin (Takara Bio, Europe AB, Göteborg, Sweden) after the 3-day activation with T cell TransAct described above, earlier shown to be a very efficient way to transduce primary T cells using retroviral vectors [22]. To facilitate transduction, 6-well suspension plates (Greiner Bio-One, Kremsmünster, Austria) were coated with RetroNectin (RN) following standard procedures as suggested by the manufacturer. For transduction, vector-containing supernatant was added in TexMACS in a total volume of 1 mL to provide a multiplicity of infection of 10. Plates were centrifuged for 1 h (1000× *g*, 4 °C) to preload vector particles onto the RetroNectin. The medium containing non-bound vector particles was removed and 1 × 10^6^ human primary T cells were added in 2 mL T-cell medium. After 24 h incubation at 37 °C, transduced T cells were harvested by carefully removing them from the plate.

### 2.4. CRISPR/Cas RNP Generation and Transfection

Primary T cells of BC#3 were activated with T cell TransAct as described above three days prior to transfection (Figure 2A). For the following KO-experiments, an additional step was added to the protocol. On day 3, after activation with TransAct, the cells were seeded on RetroNectin-coated plates. Electroporation was performed 24 h after this step as for the combined transduction protocol later.

To assemble CRISPR-Cas9-sgRNA RNP-complexes, Cas9 protein TrueCut v2 (ThermoFisher Scientific) was mixed in a PCR tube with individual sgRNAs targeting the CD45 gene. Based on preliminary experiments in T- and NK-cell lines, we chose an sgRNA directed to nucleotides 261–279 (protospacer/crRNA) with the protospacer adjacent motif (*PAM*) at nucleotides 280–282 in exon 1 of the CD45 gene: GGCTTAAACTCTTGGCATT*TGG*. The molar ratio was 1:2.5 of Cas9 to sgRNA in a total volume of approximately 2 μL. The mixture was incubated for 10 min at RT. 

After a washing step (centrifugation 310× *g*, 5 min, RT), cell concentrations were adjusted to 4 × 10^5^/15 μL OPTI-MEM (Gibco). A sample of 15 μL of cell suspension was added to the pre-built RNP-complexes, and the mixture was immediately transferred to a 1 mm cuvette avoiding generation of air bubbles. Cell-RNP mixtures were electroporated with the GenePulser Xcell (BioRad, Munich, Germany) under the following conditions: Primary T cells: 90 V, square-wave pulses, 10 ms pulse length, 3 pulses (0.1 ms pause)PM-1: 85 V, square-wave pulses, 5 ms pulse length, 3 pulses (0.1 ms pause)KHYG-1: 75 V square-wave pulses, 10 ms pulse length, 3 pulses (0.1 ms pause)

Electroporated cells were recovered from the cuvettes using their respective, pre-warmed growth medium and cultured in 1 mL in 24-well suspension plates at 37 °C. 

### 2.5. Flow Cytometry, Fluorescence-Activated Cell Sorting and Magnetic Cell Sorting 

Flow cytometry (FC) and fluorescence-activated cell sorting (FACS) were performed on a FACSCanto II and a FACS Aria IIIu (both BD Biosciences, Heidelberg, Germany), respectively. For FC and FACS, cells underwent a washing step by centrifugation at 310× *g* for 5 min at RT and were resuspended in approx. 250 μL PBS. Cells were stained with FITC- or VioBlue-labelled anti-human monoclonal REAfinity CD45 antibodies (Miltenyi Biotec) according to the manufacturer’s protocol. Unstained cells were used as controls. FC data were analyzed using FlowJo 10 (BD Biosciences).

Magnetic cell sorting (MACS) was used to separate CD45-positive (CD45^pos^) and CD45^ko^/CD19CAR-T cells three days after the CD45-KO electroporation. The cells were labeled using CD45 (TIL) MicroBeads (Miltenyi) and sorted through MACS columns into CD45^pos^ and CD45^ko^ fractions according to the manufacturer’s protocol. Successful sorting was confirmed via FC 24 h later.

### 2.6. Killing Assays

Luciferase-based assays were employed to test the killing ability of CD45^ko^/CD45CAR-T cells. Ramos and Nos-1FD4 cells were chosen as target cells after confirming homogenous CD45 surface expression via antibody staining and FC. To generate Ramos^Luc^ and Nos-1FD4^Luc^ cells, parental cells were transduced with LeGO-Luc2-iG2-Puro^+^. Pure populations of luciferase-expressing cells were obtained by culturing in the presence of 1 μg/mL puromycin (Merck). Based thereon, the viability of Luc2^+^ cells could be assessed after addition of luciferin by measuring the bioluminescence with a plate reader. For killing assays, target and effector cells were counted and seeded together in white 96-well OptiPlates (Perkin Elmer, Waltham, MA, USA). In brief, 10,000 target cells were seeded in 100 μL/well in effector cell medium, and effector cells were added at different effector-to-target (E:T) ratios to a total volume of 200 μL/ well. Co-cultured cells were incubated at 37 °C for 4 h. Thereafter, plates were centrifuged at 1000× *g* for 5 min at RT to pellet the cells. A sample of 100 μL supernatant was carefully removed with a multi-channel pipette. D-Luciferin Firefly (Biosynth, Staad, Switzerland) diluted in PBS (16 ng in 100 μL PBS) was added, and plates were incubated for 20 min in the dark. The bioluminescence intensity was analyzed with the plate reader Infinite 200 pro (Tecan, Männedorf, Switzerland) using 2000 ms integration time.

### 2.7. Statistics 

Graphs and statistical analyses were made with GraphPad PRISM (version 9) using 2-way ANOVA. *p* values below 0.05 were considered to indicate significance. Levels of significance are designated as follows: * *p* < 0.05, ** *p* < 0.01, *** *p* < 0.001, and **** *p* < 0.0001. 

## 3. Results

### 3.1. Efficient and Stable Knockout of the CD45 Protein in Primary Human T Cells

Based on preliminary studies in T and NK cell lines, we chose the sgRNA indicated in the Section 2 to knockout CD45 in primary T cells. Surprisingly, CD45-KO worked far more efficiently in primary cells than in cell lines. FC measurements after 7 days reproducibly demonstrated high KO efficiencies with an average rate of CD45-negative (CD45^ko^) cells of 51.7% (± 4.5%) (Figure 2A–C). Unexpectedly, the CD45^ko^ T cells had no growth disadvantage in culture, which resulted in stable CD45^ko^ fractions despite the presence of competing CD45^pos^ cells, as shown for cells from three different donors (buffy coats (BC) #13-#15) for 17 days after the KO (Figure 2D). In conclusion, we established an efficient method to knockout the CD45-protein in primary T cells. Along the way, we established CD45-negative PM-1 and CD45-negative KHYG-1 cell lines. Remarkably, we did not observe any impact of CD45-KO on the growth of the cell lines nor the primary T cells in vitro. 

### 3.2. Combined CD45-CAR Transduction and CD45-KO in Primary T Cells

Next, we aimed to establish a combined process for CD45-CAR transduction and CD45-KO in primary T cells. First, we transduced primary human T lymphocytes with the GALVenv-pseudotyped lentiviral CD45-CAR vector (CD45CAR_CD28c_GALV), depicted in Figure 1, following our standard protocol. In independent experiments using buffy coats from four different donors, we observed efficient transduction of 23.1–43.8% CD3 T cells. In these experiments, the gene transfer rates were determined based on BFP expression as measured on days 5–8 after transduction (Figure 3).

In order to define the optimal procedure for the combined process, we varied several parameters, particularly the order of the two modification steps and the time lags between T-cell activation, transduction, and CD45 knockout (from 24 h to 7 days). While some of the resulting protocols were impracticable (e.g., due to substantial cell death) or did not result in sufficient numbers of double-modified T cells, two modi operandi gave promising results and will in the following be referred to as protocols P1 and P2.

P1: CD45-CAR transduction—24 h resting period—CD45KO electroporation 

P2: CD45KO electroporation—3 days resting period—CD45-CAR transduction

### 3.3. CD45^ko^/CD45CAR-T Cells Mediate High-Efficient Fratricide of CD45-Positive T Cells 

As noted above, we did not observe any major impact of CD45-KO on T-cell growth characteristics. However, given the central role of CD45 in the regulation of effector cell activation, signal transduction, and cytotoxicity [14], a significant loss of function of these cells had to be expected upon knockout of CD45. In fact, previous work had shown decreased NK cell activity in CD45-negative mutants, albeit to different extents [8,9,10], whereas data on the functionality of CD45-negative T cells are largely lacking, mainly because CD45^−/−^ mice develop only very few or no T cells [23].

Contrary to the expectation, we found CD45^ko^/CD45CAR^+^ cells to mediate efficient fratricide killing of CD45^pos^ T cells as exemplified for BC#4 in Figure 4A,B. As shown in Figure 4A, using protocol P1, we obtained 41.4% CD45^ko^/CD45CAR^+^ cells (Q3 of the dot plot) three days after RNP electroporation; at this time point approx. one of six T cells (16.6%) were still CD45^hi^ (Q1 + Q2). Only 48 h later almost no CD45-positive cells were left (0.9%, Figure 4B, (Q1 + Q2)). 

These results could largely be reproduced using T cells from three different healthy blood donors (BC#13-#15); the experiment workflow is visualized in Appendix A. Initial transduction rates were 45.7%, 26.2% and 39.9% for BC#13-#15, respectively. In all cases, co-culture of CD45^ko^/CD45CAR^+^ cells with CD45^pos^ T cells resulted in complete elimination of the CD45^pos^ cells over time, albeit with slightly different kinetics (Figure 4C). In three of four experiments (BC#4, BC#13, BC#15), percentages of CD45^pos^ cells were below 10% by day 10 of the culture. Only for BC#14 did the elimination of CD45^pos^ cells take longer, but it was almost completed by day 21. One obvious reason for the slower kinetics was the lower initial CD45-KO rate obtained in this experiment. Additionally, donor-specific factors such as primary T-cell fitness (again, potentially affecting cell growth, transduction, and transfection rates) might influence the outcome of individual experiments. 

In summary, based on these experiments, we could conclude that (i) the novel CD45-CAR is functional and CD45-specific, (ii) primary human T cells survive the applied process of combined CD45-KO and retroviral CAR transduction, and (iii) CAR-T cells do not require CD45 protein on their surface for efficient effector function in vitro.

### 3.4. Efficient Killing of Ramos B Cells by CD45^ko^/CD45CAR-T Cells

To verify the activity of CD45CAR-T cells in the context of malignant targets, we tested them in luciferase-based killing assays against the B-cell leukemia line Ramos. First, we confirmed CD45-positivity of Ramos cells by CD45-antibody staining and flow cytometry; we found uniform and high CD45 expression (Appendix A). 

In the killing experiments, we tested CD45-CAR constructs with two different spacers differing in their lengths, namely CD45CAR_CD28c (CD28 derived) and CD45CAR_h23c (derived from human IgG4 CH2-CH3 domains, modified for reduced Fc receptor binding [24]). Both CD45CAR constructs facilitated efficient T-cell transduction and killing of Ramos cells, whereby the CAR construct with the shorter spacer (CD28c) was more effective in this setting. In fact, for CD45CAR_CD28c, we observed killing rates of up to 43.7% at an effector–target (E:T) ratio of 10:1 after just 4 h of co-incubation (Figure 5A). 

As described above, we tested different protocols for the production of the CD45^ko^/CD45CAR-T cells. A comparison of the results obtained with the two protocols P1 and P2 is provided in Figure 5B. In this experiment performed with BC#6, CAR-T cells generated following protocol P1 killed up to 79.8% of Ramos target cells at an E:T ratio of 5:1 during 4 h incubation. The killing activity of P2-CAR-T cells was lower, but still reached 50.4% at a 10:1 E:T ratio (Figure 5B). A summary of the numbers of CD45^ko^/CD45CAR-T cells for BC#6 obtained using the two protocols in small scale is provided in Appendix A. 

Lastly, we asked whether prolonged in vitro cultivation might impair the functional activity of CD45^ko^/CD45CAR-T. To address this question, the luciferase-based killing assay was performed on day 22 of ex vivo culture (Figure 5C). Notably, the killing of Ramos cells at that late time point was at least as efficient as 10 days earlier (compare Figure 5B). The conserved or even increased functional activity of those cultured CD45^ko^/CD45CAR-T cells might be due to the higher purity after completion of fratricide (compare Figure 4C).

In conclusion, we demonstrated the functionality of CD45^ko^/CD45CAR-T cells in several experiments with T cells from various healthy donors thus providing proof of concept for the proposed strategy. Moreover, our data led us to conclude that protocol P1 was not only the fastest method, but also resulted in the best-performing effector cells. Therefore, protocol P1 was used for further experiments. 

### 3.5. CD45^ko^/CD45CAR-T Cells Effectively Eliminate Human AML Cells 

In order to assess the lytic activity of CD45^ko^/CD45CAR-T cells against human AML cells, we made use of the novel AML cell line Nos-1FD4 recently established in our laboratory (Horn et al., manuscript in preparation). Nos-1FD4 cells show high CD45 expression as confirmed by FC (Appendix A). First, we co-incubated GFP-expressing Nos-1FD4 with CD45^ko^/CD45CAR-T cells at a 1:1 ratio. We reasoned that killing of the AML cells would be easily detectable by the disappearance of GFP^+^ cells in flow cytometry. For control, we in parallel cultured Nos-1FD4 cells in T-cell medium and did not observe any growth-inhibiting effects on Nos-1FD4 cells due to the change in culture conditions. Further, we co-cultured non-transduced T cells from the same donor with Nos-1FD4 cells to assess any potential unspecific or allo-reactive killing. In this control experiment, Nos-1FD4-GFP^+^ cells showed a clear growth advantage over the T cells dominating the cell culture already after 42 h and representing >80% of cells after 70 h, as estimated by FC (Figure 6A). In striking contrast, co-culture with CD45^ko^/CD45CAR-T cells resulted in almost complete elimination of the Nos-1FD4-GFP^+^ cells after 70 h, independent of the spacer used (Figure 6A). 

To verify the oncolytic activity of CD45^ko^/CD45CAR-T cells in an independent experimental setting, we also performed the luciferase-based killing assay as above. In the experiment shown in Figure 6B, we used CAR-T cells generated from BC#14 that were already in culture for 30 days to address the question of long-term functionality. Despite the extended in vitro culture, the CD45^ko^/CD45CAR-T cells still mediated very effective killing of up to 60% of the leukemic cells in the short-term (4 h) luciferase assay, confirming the above results obtained with Ramos target cells. 

In summary, CD45^ko^/CD45CAR-T cells were able to survive and effectively kill two different CD45-positive leukemia cell lines. Despite the CD45-KO, their functional activity remained largely preserved even after long-term ex vivo culture for >4 weeks. 

### 3.6. No Negative Impact of CD45-KO on Growth and Effector Functions of CAR-T Cells In Vitro

So far, we have shown that CD45^ko^/CD45CAR-T cells can efficiently be generated, expanded, and used in killing assays. However, by using the CD45CAR, we could not completely exclude the previously suggested negative effects of CD45-KO on CAR-T cell growth and function, since any presence of CD45 on the T cells did result in fratricide, i.e., a selective advantage of the CD45^ko^ cells.

To overcome this limitation, we used a standard CD19-CAR for the following experiments. CD19CAR-T cells were produced using the T cells of three different healthy donors (BC#21-#23). CD19-CAR transduction and CD45-KO followed protocol P1. To obtain pure populations, CD45^ko^ and CD45^pos^ cells were separated by MACS. 

Addressing both the proliferation and the effector functions of CD45^ko^/CD19CAR-T as compared to CD45^pos^/CD19CAR-T cells, we in parallel co-cultured both effector cell types with CD19-positive Ramos-GFP^+^ cells starting at an E:T ratio of 1:5 and monitored CAR-T cell numbers for 1 week (Figure 7A). Remarkably, even though we observed very diverse proliferation rates between CD19CAR-T cells derived from the three different BCs, there were essentially no differences between CD45^ko^/CD19CAR-T and CD45^pos^/CD19CAR-T cells for a given donor (Figure 7A). To assess the oncolytic potential of the different CAR-T cells, we quantified the numbers of Ramos-GFP^+^ cells over time in the co-culture. In Figure 7B, the numbers of Ramos-GFP^+^ cells were normalized in the individual cultures to allow the comparison of killing kinetics with effectors from the three different donors. Despite the low initial E:T ratio of 1:5, we saw the efficient elimination of tumor cells after 7 days (Figure 7B), again without significant differences between CD45^ko^ and CD45^pos^ CAR-T cells. This finding was confirmed in a direct short-term (4 h) luciferase-based killing assay on day 30 (Figure 7C).

In conclusion, the direct comparison of CD45^ko^ and CD45^pos^ CAR-T cells indicates that the absence of CD45 has no negative impact on the growth behavior and functional activity of CAR-T cells in vitro. 

### 3.7. CD45^ko^-CD19-CAR-T Cells Mediate Multiple Killing Rounds In Vitro

In a final experiment, we addressed the perseverance of CD45^ko^ CAR-T cells in vitro, and particularly their capability to be repetitively activated and exert cytotoxic activity. To do so, after transduction and knockout, we sorted CD19CAR T cells into CD45^pos^ and CD45^ko^ cells. Both populations were re-stimulated with either TransAct (following the manufacturer’s protocol) or CD19^pos^/GFP-transgenic Ramos cells (Figure 8A). For the first re-stimulation, we used Ramos cells at an E:T ratio of 5:1. However, at this ratio, the Ramos cells were rapidly eliminated as detected by FC after 48 h. Therefore, we used an excess of Ramos cells for the second re-stimulation (E:T ratio of 1:5). This re-stimulation provided a strong proliferation signal to both CD45^pos^ and CD45^ko^ CD19CAR-T cells, which grew faster than TransAct re-stimulated cells (Figure 8B). Moreover, re-stimulation with Ramos cells, but not with TransAct, resulted in a substantial increase in CD19CAR^pos^ T cells, independent of CD45 expression (Figure 8C). Despite the low starting E:T ratio, GFP-positive Ramos cells (Figure 8D, insert) were essentially completely eliminated within 72 h of co-culture as demonstrated by FC (Figure 8D). At the end of the co-culture, we used re-stimulated CD45^pos^ and CD45^ko^ CD19CAR-T cells in our standard luciferase-based 4 h killing assay. As shown in Figure 8E, repeatedly stimulated CD19CAR-T cells showed strongly increased cytotoxic activity with approx. 90% killing rates within just 4 h of incubation (for comparison see Figure 7C). FC analysis indicated no difference in the expression of exhaustion markers PD-1 and TIM3 between CD45^ko^ and CD45^pos^ cells at the end of culture (Figure 8F). Together, these data prove that CD45^ko^ CAR-T cells can perform multiple rounds of killing and show no exhaustion phenotype in vitro.

## 4. Discussion

Data on the functionality of CD45^ko^ immune effector cells are scarce. Early work with the RNK-16 cell line (rat) revealed that wild-type cells, but not CD45-negative mutants were cytotoxically active [25]. In contrast, it was later shown in the mouse model that CD45-negative primary NK cells exhibited defects in activation, signal transduction and cytokine production, whereas cytotoxic activity was preserved in vitro, albeit to a reduced extent [26,27]. However, the observed residual functionality of CD45-negative primary NK cells in the mouse might not be predictive for human cells, since activation and signal transduction in murine and human NK cells are in part different. For example, murine and human NKG2D differ significantly in structure and the signaling pathways used [28]. Moreover, binding and “crosslinking” with anti-CD45 antibodies have been found to inhibit the cytotoxic effect of primary human NK cells, but not primary human T cells, although the exact mechanism remains to be elucidated. The antibodies could stimulate a cytotoxic-responsive function of CD45 or trigger an inhibitory effect [29,30,31]. 

Insights into the functionality of CD45-negative T cells are even more lacking, mainly because CD45^−/−^ mice develop only very few or no T cells [23]. Indeed, the role and relevance of CD45 does not only differ considerably between species, but also between T and NK cells [32]. Recent data point to a “gatekeeper” function of CD45 in T cells, which enables graded signaling outputs while filtering weak or spurious signaling events [33]. CAR-mediated T-cell activation was reported to be dependent on the size difference between the CAR-antigen pair and CD45 [34]. It is, therefore, unknown how CD45 knockout would affect T-cell effector function in terms of activation, proliferation, and cytotoxicity and even more so, when signaling occurs through a CAR.

In this study, we generated CD45^ko^/CD45CAR-T and CD45^ko^/CD19CAR-T cells and studied their functionality in vitro. The protocol established by us allows for efficient knockout of CD45 in primary T cells, enabling the production of CD45^ko^CAR-T cells against different antigens, most relevantly CD45^ko^/CD45CAR-T cells.

In the co-culture of CD45^pos^ and CD45^ko^ T cells, the relative proportion of the CD45-negative fraction was maintained over the total culture period of approx. three weeks. These data indicate that in vitro proliferation and survival of T lymphocytes were not affected by the CD45 knockout. Upon CD45-CAR transduction, residual CD45-positive cells were rapidly eliminated by fratricide, demonstrating the potent cytotoxicity of the CD45^ko^/CD45CAR-T cells in vitro. Using various in vitro assays, we further observed high cytotoxic activity of CD45^ko^/CD45CAR-T cells against malignant cell lines Ramos and Nos-1FD4. The pronounced cytotoxicity of CD45^ko^/CD45CAR-T cells was preserved even after four weeks of in vitro culture; a time frame substantially exceeding that for current large-scale production of commercially available CAR-T products. 

Even though we observed pronounced functional activity of CD45^ko^/CD45CAR-T cells, we could not fully exclude a potential negative impact of the knockout on their performance. Direct comparison with CD45-positive CD45CAR-T cells would not be meaningful, since in those cells the CAR-triggered effector function is compromised by fratricide. Therefore, we used an established CAR against CD19, to compare the CAR-triggered effector function of CD45^pos^ and CD45^ko^/CD19CAR-T cells. We found a very similar effector functionality of CD45^ko^/CD19CAR-T cells compared to CD45^pos^/CD19CAR-T cells in terms of proliferation and cytotoxicity. Using an in vitro re-challenge model with CD19-positive Ramos cells we also observed that CD45^ko^/CD19CAR-T cells had the same capability for serial re-stimulation and killing activity as CD45^pos^/CD19CAR-T cells.

A recent study published very shortly before our first submission reported similar findings including efficient short-term cytotoxicity in vitro but impaired long-term cytotoxicity in an in vivo mouse model [35]. In our study, we provide data for the first time on the functionality of CD45^ko^ CAR-T cells after extended in vitro culture. Two different CAR readout systems (CD19 and CD45) indicated essentially identical activation, proliferation, and cytotoxic activities of CD45^ko^ and CD45^pos^ CD45CAR-T cells against Ramos cells over relatively long periods of in vitro culture, including the ability to perform multiple killings. In addition, CD45^ko^/CD45CAR-T cells mediated the efficient killing of AML cells even after four weeks of in vitro culture. 

In order to further translate the proposed principle, it will obviously be essential to confirm the observed pronounced in vitro activity in suitable in vivo models. In their study, Wellhausen et al. [35] reported the inability of CD45^ko^ T cells equipped with a second-generation CD19-CAR to maintain long-term antitumor efficacy in NOD-SCID-IL2rγ−/− (NSG) mice, which was in striking contrast to CD45-epitope edited CAR-T cells. Unfortunately, they did not provide data on the activity of their CD45^ko^ CAR T cells after extended in vitro culture [35]. In this context, it could be relevant that contrary to their work we used a third-generation CAR construct. Third-generation CARs were previously shown to facilitate improved in vivo persistence/functionality [36,37,38], which might have contributed to the pronounced long-term activity observed in our in vitro models. To address this hypothesis, detailed analyses of signaling pathways will be required in subsequent studies.

Since CD45 is present on all nucleated hematopoietic cells, the clinical application of CD45-directed CAR-T cells is expected to be associated with broad on-target-off-tumor toxicity. Consequently, its use would require the rescue of the hematopoietic system by an autologous or allogeneic stem cell transplantation, which is a common procedure in high risk or refractory/relapsed blood cancers. In this context, the activity of the CD45^ko^/CD45CAR-T cells will need to be terminated prior to stem cell infusion and thus limited potent short-term activity may not necessarily be a disadvantage. Alternatively, the termination or suppression of CAR-T activity could be achieved by transient expression of the CAR, introduction of one of several safety switches under investigation [39] or immunosuppressive conditioning agents like T-cell targeting antibodies. Finally, CD45 epitope editing [35] in the stem cell graft to protect the new hematopoiesis from the CD45-directed CAR-T cells might also be an opportunity.

## 5. Conclusions

We have reported the efficient generation of CD45^ko^/CD45CAR-T and CD45^ko^/CD19CAR-T cells with high functional activity in vitro. While our data provide important proof-of-concept for the suitability of CD45 as a CAR-T target and the functionality of CD45^ko^ CAR-T cells, it obviously will need confirmation in suitable animal models before potential translation into clinical testing. Importantly, in a parallel work Wellhausen et al. [35] has already provided strong in vivo evidence for the applicability of CD45-directed CARs for universal blood cancer immunotherapy using CD45 epitope base editing. 

## Figures and Tables

**Figure 1 cancers-16-00334-f001:**
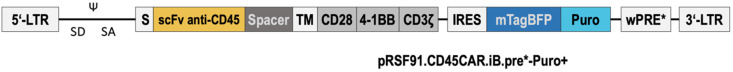
Schematic of the γ-retroviral CD45CAR vector with the short CD28c spacer. The SF91 based γ-retroviral vector backbone pRSF91.iB.pre* [19] with the safety-optimized wPRE* [20] was chosen to facilitate efficient CAR expression. LTR = long terminal repeat; SD = splice donor; SA = splice acceptor, S = signal peptide; scFv = single chain variable fragment; TM = transmembrane; CD28, 4-1BB and CD3ζ = intracellular signal transduction elements for T-cell activation; IRES = internal ribosome entry site derived from encephalomyocarditis virus; mTagBFP = monomeric blue fluorescent protein; puro = puromycin resistance; wPRE = woodchuck hepatitis virus post-transcriptional response element. Not to scale.

**Figure 2 cancers-16-00334-f002:**
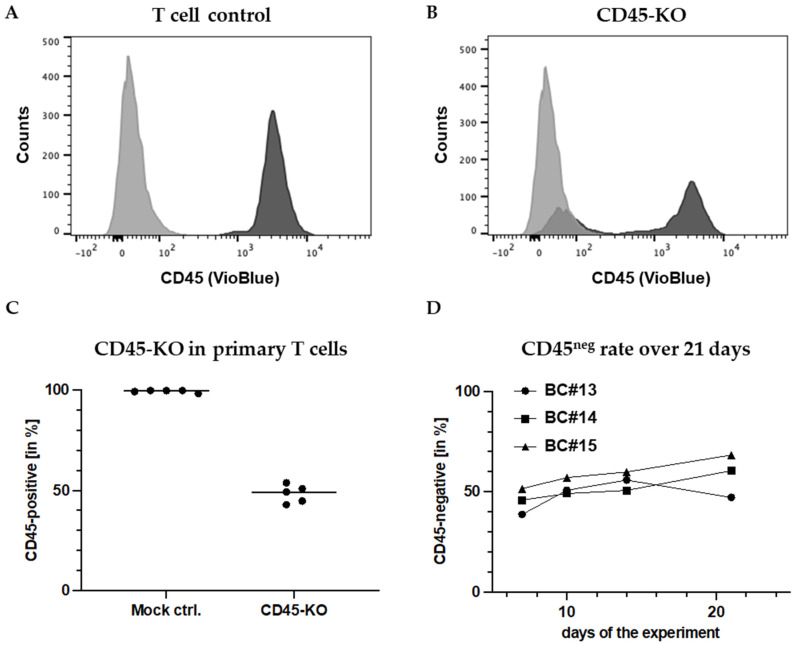
High-rate CD45 knockout in primary human T cells. FC measurement of CD45 expression on (**A**) mock (electroporated without RNPs) and (**B**) primary human T cells (BC#3) transfected with CRISPR/Cas9-RNPs to knockout CD45, seven days after electroporation. (**C**) Reproducibly high CD45-KO rates in primary T cells from different donors. As mock control cells, T cells were electroporated without RNPs. (**D**) CD45-KO rates remain stable in primary T cell cultures after the KO-electroporation for at least 2.5 weeks.

**Figure 3 cancers-16-00334-f003:**
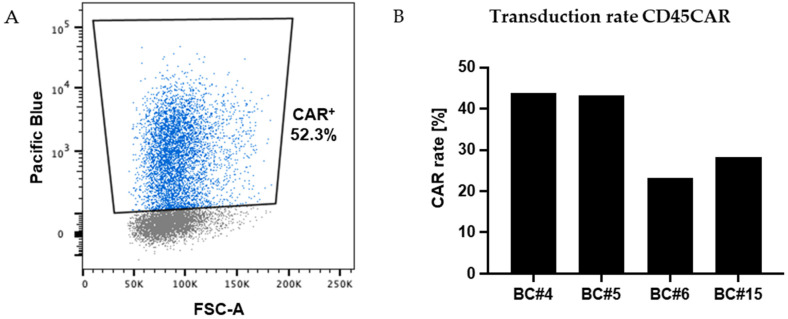
Efficient transduction of primary human T cells. (**A**) FC analysis revealed high level expression of the marker gene BFP in a relevant proportion of primary T cells as exemplarily shown for BC#4 at day 3 post transduction. (**B**) Transduction rates for T cells from four different donors as assessed by FC based on BFP expression 5–8 days post transduction.

**Figure 4 cancers-16-00334-f004:**
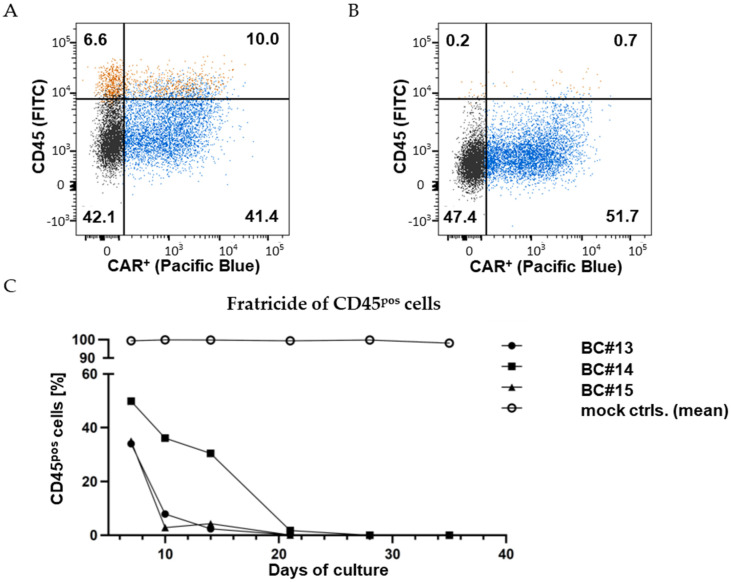
CD45CAR-T cells mediate fratricide. FC measurement 3 days (**A**) and 5 days (**B**) after the CD45-KO electroporation of primary CD45CAR-T cells (BC#4). The cells were stained with a CD45-antibody (FITC). Vector-mediated transgene expression was quantified based on detection of the IRES-driven BFP. Relative cell numbers (%) in the four quadrants are indicated. (**C**) Complete fratricide of CD45^pos^ T cells in three independent experiments by day 28. CD45^ko^/CD45CAR-T cells were produced performing protocol P1.

**Figure 5 cancers-16-00334-f005:**
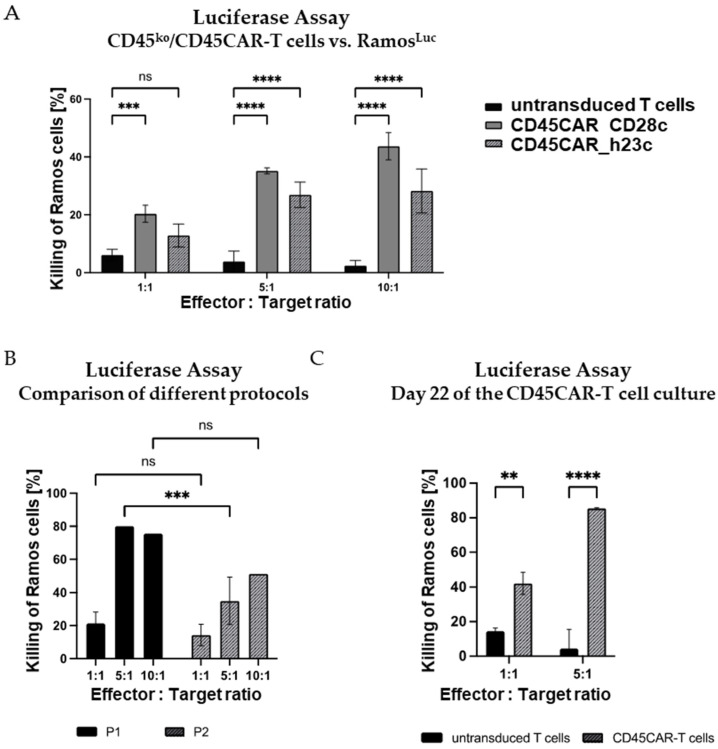
Efficient killing of Ramos B cells by CD45^ko^/CD45CAR-T cells within 4 h. (**A**) CD45^ko^/CD45CAR-T cells are functional with both the long and the short spacers. Transduction and electroporation of T cells from BC#4 were performed following protocol P1. The 4 h killing assays (*n* = 3) were carried out on day 10 of the experiment. (**B**) Comparison of killing activities of CD45^ko^/CD45CAR-T cells produced with the two different protocols P1 and P2. T cells were from BC#6, the killing assays (*n* = 1–3) were carried out on day 12 after T-cell activation. Only the short spacer was used in this experiment. (**C**) High killing activity of CD45^ko^/CD45CAR T cells after prolonged in vitro culture. CAR-T cells were produced following protocol P2 using BC#5. The killing assays (*n* = 2–3) were performed on culture day 22. Levels of significance are designated as follows: ** *p* < 0.01, *** *p* < 0.001, and **** *p* < 0.0001, ns = not significant.

**Figure 6 cancers-16-00334-f006:**
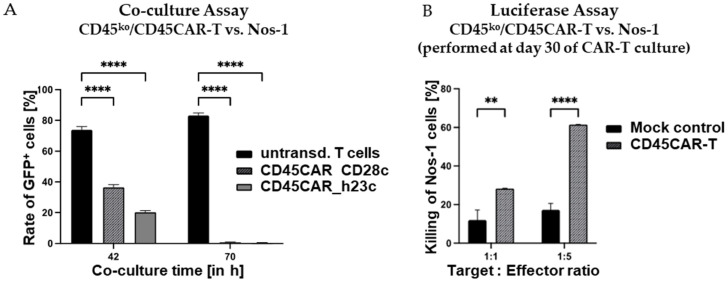
Pronounced oncolytic activity of CD45^ko^/CD45CAR-T cells against Nos-1FD4 AML cells. (**A**) Co-culture of CD45^ko^/CD45CAR-T cells (BC#4) against Nos-1FD4-GFP^+^ AML cells. Effector and target cells were cultivated at a 1:1 ratio and percentages of Nos-1FD4-GFP^+^ cells were monitored by flow cytometry based on GFP expression. As evident (left bars), Nos-1FD4 cells had a growth advantage over non-transduced T cells. In contrast, in the presence of CD45^ko^/CD45CAR-T cells, Nos-1FD4 cells essentially completely disappeared after 70 h of co-culture (*p* < 0.0001 for both CAR constructs and both time points, *n* = 3). (**B**) Standard 4 h luciferase-based killing assays using growth-factor independent Nos-1FD4-ITD^Luc^ cells as targets. CD45^ko^/CD45CAR-T cells (BC#14) had been cultivated for 30 days before their use in the killing assays (*n* = 2 for CAR-T and *n* = 3 for mock). Levels of significance are indicated as follows: ** *p* < 0.01 and **** *p* < 0.0001.

**Figure 7 cancers-16-00334-f007:**
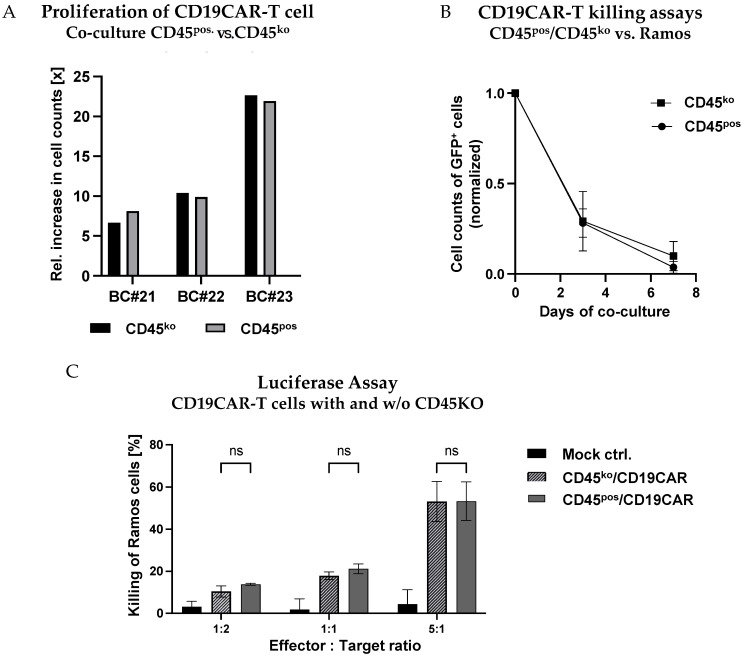
No negative impact of CD45-KO on CD19CAR-T cell proliferation and functional activity. (**A**,**B**) CD45^ko^ and CD45^pos^/CD19CAR-T cells were co-cultured with Ramos-GFP^+^ cells at an E:T ratio of 1:5. (**A**) No major differences in numbers of cell doublings for CD45^ko^ and CD45^pos^/CD19CAR-T cells derived from three different BCs (BC#21-BC#23) after 1 week of co-culture. (**B**) Both CD45^ko^ and CD45^pos^/CD19CAR-T cells mediate efficient elimination of Ramos-GFP^+^ cells in co-culture. Relative Ramos cell numbers were quantified by flow cytometry based on GFP expression, absolute numbers were counted at days 3 and 7 of the co-culture (three different BCs). (**C**) Luciferase-based killing assays with Ramos-Luc-GFP^+^ cells as targets cells and an incubation time of 4 h (*n* = 3 different BCs, except CD45^pos^/CD19CAR-T cells at E:T ratio of 1:1 [*n* = 2]).

**Figure 8 cancers-16-00334-f008:**
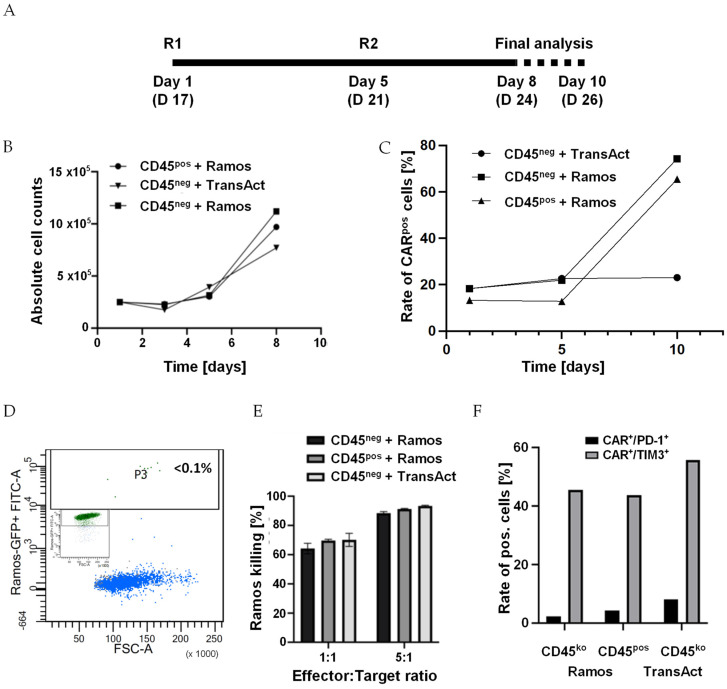
CD45ko CAR-T cells are repeatedly re-stimulated by CD45^pos^ malignant cells. (**A**) Sorted CD45^ko^ and CD45^pos^ CD19 CAR-T cells were re-stimulated [R] on days 17 (new day 1) and 21 (5) of culture. Final analyses were performed by counting and FC on days 8 and 10. (**B**) Direct comparison of growth kinetics of CD45^ko^ and CD45^pos^ CD19 CAR-T cells after re-stimulation with CD19^pos^ Ramos cells on days 1 and 5. First re-stimulation with Ramos cells was performed at an E:T ratio of 5:1, second re-stimulation at an E:T ratio of 1:5. For control, CD45^ko^/CD19CAR-T cells were also re-stimulated with TransAct. (**C**) Increased CAR expression after re-stimulation of both CD45^ko^ and CD45^pos^ CD19-CAR T cells with Ramos cells, but not with TransAct. (**D**) FC analysis of GFP^pos^ Ramos cells before (inset) and after 3 days co-culture with CD45^ko^/CD19CAR-T cells (E:T ratio 1:5). (**E**) Re-stimulated CD45^ko^/CD19CAR-T and CD45^pos^/CD19CAR-T cells mediate efficient killing in the 4 h killing assay (3 technical replicates, E:T ratios normalized based on CAR expression. (**F**) Expression of exhaustion markers PD-1 and TIM3 on CAR-T cells in the different groups at final analysis.

## Data Availability

All data associated with this study are present in the paper or the Appendix A. All raw data are available upon request.

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
