# Peer review of "CD45-Directed CAR-T Cells with CD45 Knockout Efficiently Kill Myeloid Leukemia and Lymphoma Cells In Vitro Even after Extended Culture"

_cancers, 2024, doi:10.3390/cancers16020334_

Round 1

Reviewer 1 Report

Comments and Suggestions for Authors

Harfmann and colleagues report on CD45 knockout model for CAR-T

This article is very innovative and very much actual. The implications of these finding have a strong clinical relevance. I believe that it should be published in its current format. 

Author Response

We thank the reviewer for this very positive feedback.

Reviewer 2 Report

Comments and Suggestions for Authors

In their manuscript, Harfmann et al. attempted to design CD45-targeted CAR T cells with an endogenous knockout of CD45 in order to minimize the adverse effects of feticide. The manuscript is well-written and thoroughly documented.

CD45 is a cell surface antigen in all immune and hematopoietic cells, including HSCs and HSPCs. Thus, targeting CD45 is particularly suitable for a conditioning regimen before HSCT. Recently, there has been a trend in targeting CD45 with CART cells to eradicate blood cancer. In a very recent study by Nils Wellhausen et al,(PMID: 37651540) it was shown that eliminating CD45 from CAR 45 T cells did not affect their cytotoxicity, granulation, or cytokine release in vitro, but it did affect the persistence and expansion of CD45KO CAR45 T cells in vivo.

However, the current manuscript only establishes the same in vitro findings without any in vivo data. Therefore, the authors should discuss the novelty of their findings in relation to the previously mentioned work. It is important to note that this study sheds light on the potential benefits of CD45KO CAR45 T cells, and further research is needed to explore the efficacy of this strategy in vivo.

Comments on the Quality of English Language

Minor spelling check requested.

Author Response

In their manuscript, Harfmann et al. attempted to design CD45-targeted CAR T cells with an endogenous knockout of CD45 in order to minimize the adverse effects of feticide. The manuscript is well-written and thoroughly documented.

We thank the reviewer for this positive feedback.

CD45 is a cell surface antigen in all immune and hematopoietic cells, including HSCs and HSPCs. Thus, targeting CD45 is particularly suitable for a conditioning regimen before HSCT. Recently, there has been a trend in targeting CD45 with CART cells to eradicate blood cancer. In a very recent study by Nils Wellhausen et al,(PMID: 37651540) it was shown that eliminating CD45 from CAR 45 T cells did not affect their cytotoxicity, granulation, or cytokine release in vitro, but it did affect the persistence and expansion of CD45KO CAR45 T cells in vivo.

However, the current manuscript only establishes the same in vitro findings without any in vivo data. Therefore, the authors should discuss the novelty of their findings in relation to the previously mentioned work. It is important to note that this study sheds light on the potential benefits of CD45KO CAR45 T cells, and further research is needed to explore the efficacy of this strategy in vivo.

We thank the reviewer for this comment. Two paragraphs of the Discussion section are dedicated to the discussion of our findings vis-à-vis the study by Wellhausen et al.

The main differences include the use of a third-generation CAR in our study (in contrast to a second-generation CAR used by Wellhausen et al) and the extended in-vitro culture. We hope the newly included statements in the Discussion help highlight this potentially important aspect.

A recent study published very shortly before our first submission reported similar findings including efficient short-term cytotoxicity in vitro but impaired long-term cytotoxicity in an in-vivo mouse model [35]. In our study we for the first time provide data on functionality of CD45ko CAR-T cells after extended in-vitro culture. Two different CAR readout systems (CD19 and CD45) indicated essentially identical activation, proliferation, and cytotoxic activity of CD45ko and CD45pos CD45CAR-T cells against Ramos cells over relatively long periods of in-vitro culture, including the ability to perform multiple killings. In addition, CD45ko/CD45CAR-T cells mediated efficient killing of AML cells even after four weeks of in-vitro culture.

In order to further translate the proposed principle, it will obviously be essential to confirm the observed pronounced in-vitro activity in suitable in-vivo models. In their study, Wellhausen et al. [35] reported the inability of CD45ko T cells equipped with a second-generation CD19-CAR to maintain long-term antitumor efficacy in NOD-SCID-IL2rγ−/− (NSG) mice, which was in striking contrast to CD45-epitope edited CAR-T cells. Unfortunately, they did provide data on the activity of their CD45ko CAR T cells after extended in-vitro culture [35]. In this context it could be relevant that contrary to their work we used a third-generation CAR construct. Third-generation CARs were previously shown to facilitate improved in-vivo persistence/ functionality [36-38], which might have contributed to the pronounced long-term activity observed in our in-vitro models. To address this hypothesis, detailed analyses of signaling pathways will be required in subsequent studies.

Comments on the Quality of English Language

Minor spelling check requested.

We are grateful for this request. The whole manuscript has been thoroughly read and edited by a native speaker. Please note that minor corrections have not been highlighted in the revised version.

Reviewer 3 Report

Comments and Suggestions for Authors

The authors here tried to show the efficient production of highly and durably active CD45ko/CAR-T cells and showed CD45 knockout did not impair functionality of CAR-T cells in vitro. However, the authors need to work further on this study to make the manuscript publishable in this journal.

1. The authors need to show CD45 as the universal antigen in the hematological cancers that can be targeted with CD45 knockout CAR T cell.

2. The authors showed work to show the potent activity against multiple hematologic cancer cell lines, include multiple cell line and primary AML xenografts.

3. Manuscript figures needs better resolution and better representation and legends needs to re-arrange.

4. Minor English language, Scientific language edits and fonts needs modification.

Comments on the Quality of English Language

Minor English language

Author Response

The authors here tried to show the efficient production of highly and durably active CD45ko/CAR-T cells and showed CD45 knockout did not impair functionality of CAR-T cells in vitro. However, the authors need to work further on this study to make the manuscript publishable in this journal.

We thank the reviewer for this feedback. We have addressed the following points:

  1. The authors need to show CD45 as the universal antigen in the hematological cancers that can be targeted with CD45 knockout CAR T cell.

We thank the reviewer for this comment. We do agree that CD45 is expressed on nearly all hematopoietic cells. Taking into account the increasing availability of CAR-T cells targeting lineage-specific antigens such as CD19, we believe the place for CD45-targeted therapy will mainly be in myeloid disease as well as conditioning prior to autologous or allogeneic stem cell transplantation. In the introduction section we touch on the possible broad application of CD45-CAR-T cells:

CD45 is a type-I transmembrane protein with various isoforms found on almost all hematopoietic cells (except erythrocytes and platelets) and a regulator of signaling thresholds in immune cells [9]. It is expressed on 85-90% of ALL and almost all AML cells but not on non-hematopoietic tissues [10,11]{Andres, 1983 #588;Omary, 1980 #589}. Targeting CD45 with CAR-based cell therapy may therefore be in principle employed for treatment of both lymphoid and myeloid hematological diseases.

  1. The authors showed work to show the potent activity against multiple hematologic cancer cell lines, include multiple cell line and primary AML xenografts.

In our study, we have shown efficient elimination of different target cells by CD45ko CAR-T cells, namely primary human T cells, Ramos lymphoma cells and Nos1 AML cells. Obviously, we agree with the reviewer, that in the future further studies will need to be done to affirm the potential of CD45ko CAR-effector cells. Our study aims to provide a proof-of-concept for the efficient production and cytotoxic potential of CD45ko CAR-T cells, even after protracted in-vitro culture periods. We believe our study shows promising data and lays the ground work for further studies including xenograft models and hopefully clinical trials.

  1. Manuscript figures needs better resolution and better representation and legends needs to re-arrange.

We have modified the figures to improve legibility. We have checked and re-arranged the legends to improve clarity

  1. Minor English language, Scientific language edits and fonts needs modification.

We are grateful for this request. The whole manuscript has been thoroughly read and edited by a native speaker. Please note that minor corrections have not been highlighted in the revised version.